# Socioeconomic Deprivation, Sleep Duration, and Mental Health during the First Year of the COVID-19 Pandemic

**DOI:** 10.3390/ijerph192114367

**Published:** 2022-11-03

**Authors:** Stephanie Griggs, Christine Horvat Davey, Quiana Howard, Grant Pignatiello, Deepesh Duwadi

**Affiliations:** 1Frances Payne Bolton School of Nursing, Case Western Reserve University, Cleveland, OH 44106, USA; 2Department of Physiology and Biophysics, Case Western Reserve University, Cleveland, OH 44106, USA

**Keywords:** COVID-19, sleep duration, mental health, socioeconomic deprivation, social deprivation index, health outcomes

## Abstract

The coronavirus disease 2019 (COVID-19) has had a rapid and sustained negative impact on sleep and mental health in the United States with disproportionate morbidity and mortality among socioeconomically deprived populations. We used multivariable and logistic regression to evaluate the associations among sleep duration, mental health, and socioeconomic deprivation (social deprivation index) in 14,676 Ohio residents from 1101 zip code tabulation areas from the 2020 Behavioral Risk Factor Surveillance System (BRFSS) survey. Higher socioeconomic deprivation was associated with shorter sleep and poorer mental health after adjusting for covariates (age, sex, race, education, income, and body mass index) in the multivariable linear regression models. Those in the highest socioeconomically deprived areas had 1.6 and 1.5 times higher odds of short sleep (duration < 6 h) and poor mental health (>14 poor mental health days), respectively, in the logistic regression models. Previous researchers have focused on limited socio-environmental factors such as crowding and income. We examined the role of a composite area based measure of socioeconomic deprivation in sleep duration and mental health during the first year of COVID-19. Our results suggest the need for a broader framework to understand the associations among socioeconomic deprivation, sleep duration, and mental health during a catastrophic event.

## 1. Introduction

The coronavirus disease 2019 (COVID-19) pandemic has underscored health disparities in the United States. For example, neighborhoods with lower median family income and higher unemployment rates have higher rates of COVID-19 transmission, hospitalization, and death [1,2,3,4]. There is an established relationship between poor health outcomes and socioeconomic deprivation, often assessed via the social deprivation index (SDI), a composite measure of the effects of poverty and deprivation based on income, education, employment, housing, household characteristics, transportation, and demographics [5]. The SDI permits the assessment of the individual components to understand the nuanced population effects or socio-economic variability in health outcomes of social determinants of health. The COVID-19 pandemic upended the delicate balance between sleep and mental health [6]. Evaluating the role of socioeconomic deprivation in sleep and mental health is important to (1) highlight inequality in highly overlooked domains of health behaviors, (2) understand how sleep and mental health are influenced, and (3) target higher-risk groups and areas for future interventions.

Social determinants of health are the social and economic conditions that impact a vast array of health and quality of life outcomes, including sleep and mental health [7]. Social determinants of health are classified into five main areas including healthcare access and quality, education, social and community context, neighborhood and built environment, and economic stability [7]. COVID-19 outcomes have been profoundly impacted by social determinants of health. For example, homelessness is linked to increased risk of COVID-19 transmission due to crowded living spaces and decreased access to testing facilities [8]. The effects of social determinants of health are far-reaching and have been magnified by the COVID-19 pandemic. Understanding the socioeconomic deprivation within the context of the COVID-19 pandemic can provide important insights into sleep and mental health outcomes.

The National Sleep Foundation recommends that adults sleep 7–9 h per night, and about a third of Americans report insufficient sleep (defined as <7 h) [9]. Sleep of insufficient quality or quantity is associated with higher body mass index (BMI), impaired body weight regulation (lower leptin and higher ghrelin levels), poorer insulin sensitivity, and lower daytime alertness and has been linked to multiple chronic conditions including overweight/obesity, type 2 diabetes, cardiovascular disease, stroke, dementia, and cancer [10,11,12,13,14]. During the COVID-19 pandemic years 2019 to 2021, sleep disturbances affected approximately 41% with higher rates during the lockdown compared to no lockdown, 42.49% versus 37.97% in a recent systematic review and meta-analysis of 250 studies (493,475 participants) across 49 countries [15]. However, there are few studies where the impact of the COVID-19 pandemic on sleep sufficiency has been examined. The COVID-19 impact has varied among populations with individuals reporting adequate sleep duration pre-pandemic and shorter sleep duration and increased sleep complaints during the pandemic [16,17,18]. The effect of sleep behavior on health outcomes has increasingly been studied with an emerging literature on social determinants of sleep [9,19,20,21]. For instance, lower socioeconomic status (individual level) and socioeconomic deprivation (area level) are associated with short or long sleep duration along with sleep disorders (e.g., sleep apnea) which all relate to increased mortality risk [22,23,24,25,26]. The COVID-19 pandemic has also magnified preexisting sleep disparities across individuals [6,27,28,29,30,31]. However, less is known about the association between neighborhood socioeconomic characteristics and the nuance of urban-rural classifications on sleep. Further, compared to those with sufficient sleep, individuals with insufficient sleep have nearly threefold increased odds of mental distress [32].

There is increasing concern about the impact of the pandemic on mental health [33,34]. Those with limited economic and social resources are at a higher risk for these impacts [35]. There were considerable increases in anxiety, depressive disorders, and suicidal ideation in mid-March–February 2021 (during the pandemic) compared to the same period in 2019 (pre-pandemic) in the United States [36]. In another national United States survey of 790,633 individuals, there was a significant increase in the percentage of adults with anxiety or depressive disorder symptoms (36.4% to 41.5%) during the past 7 days, as well as an increased need for mental health counseling or therapy during the past 4 weeks (from 9.2% to 11.7%) [37,38]. The higher need for mental health counseling persisted after the height of the COVID-19 pandemic in the United States with an increase in the percentage of adults needing treatment within the past 12 months (19.2% to 21.6%) [37,38].

The purpose of this study was to quantify levels of socioeconomic deprivation and examine associations among socioeconomic deprivation, sleep duration, and mental health while adjusting for covariates (e.g., age, sex at birth, racial identity, education, body mass index) during the first year of the COVID-19 pandemic. Specifically, through a cross-sectional investigation of Ohio residents from the Behavioral Risk Factor Surveillance System (BRFSS) in 2020, we tested the following hypotheses: (1) higher socioeconomic deprivation would be associated with shorter sleep and poorer mental health and (2) those in the highest socioeconomically deprived areas would have higher odds of short sleep (defined as <6 h per night) and poor mental health (defined as >14 days of poor mental health per month).

## 2. Materials and Methods

### 2.1. Data Sources and Study Population

The Ohio BRFSS 2020 data were used for this analysis. The BRFSS is a random-digit dialed state-based telephone survey about health practices, conditions, and risk among non-institutionalized adults ages 18 years and older conducted by the Ohio Department of Health and supported by the Centers for Disease Control and Prevention (CDC). Data are collected from a random probability sample representative of the 14 geographic regions in Ohio. The corresponding author’s institution, Case Western Reserve University approved the study (IRB # STUDY20221217). Case Western Reserve University has a federal-wide assurance.

### 2.2. Variables and Measures

Individual demographic variables of interest included age, sex at birth, race/ethnicity, and education. Covariates were selected based on a priori knowledge and included age, biological sex, and race/ethnicity, education, and body mass index. An area-level variable the Social Deprivation Index (SDI) was generated from the zip codes. The SDI is a composite measure of six domains collected in the American Community Survey: income (percent living in poverty), education (percent with less than 12 years of education), housing (percent living in rented housing unit and percent living in overcrowded housing unit), household characteristics (percent single-parent households with dependents <18 years), transportation (percent of households without a car), and employment (percent non-employed adults under 65 years of age) [5]. The SDI is an estimate of health care access and health outcomes within a rational primary care service area based on zip code tabulation (ZCTA). The SDI was updated with data from 5-year estimates from 2011–2015 [5]. We used ZCTA’s in this study based on generalized U.S. Postal Service Zip Codes [5].

Self-reported sleep duration was a single-item measure. Respondents were asked “on average, how many hours of sleep do you get in a 24 h period?”. Responses range from 1–24 h. We excluded implausible values for sleep duration (>18 h). The significance was not altered with the removal of implausible values. We used hours in the linear regression models. For the logistic regression models, we defined short sleep as ≤6 h consistent with other studies on sleep duration and the National Sleep Foundation Recommendation [39].

Mental health was derived from a single item measure. Respondents were asked “Now thinking about your mental health, which includes stress, depression, and problems with emotions, for how many days during the past 30 days was your mental health not good?”. We used poor mental health days from 0 to 30 in the linear regression models. For logistic regression, poor mental health was defined as 14 or more days of poor mental health in the past 30 days. This approach for defining poor mental health days has been documented in previous research [40,41].

### 2.3. Statistical Analysis

A quantitative descriptive approach was used to characterize socioeconomic deprivation, sleep duration, and mental health among the individuals in the study. Bivariate correlations, multivariable linear, and multivariable logistic regression models were used to examine the relationships among socioeconomic deprivation, sleep duration, and mental health. 

To evaluate explanatory contributions of socioeconomic deprivation to sleep duration and mental health, we performed a series of multivariable linear regression models. To assess exposure in the logistic regression model, overall SDI was dichotomized into two groups (10% most socially deprived and the remaining 90% as a comparator group). This approach for determining risk in those most socially deprived, the highest 10% of SDI scores, has been documented in previous research [42]. We defined short sleep as ≤6 h and poor mental health as 14 days or more in the past 30 days consistent with prior research and national recommendations [39,40,41]. Statistical significance was set at *p* < 0.05.

## 3. Results

### 3.1. Sample Characteristics

A population of 14,676 individuals aged 18–99 (mean age 54.9 ± 18.1 years) residing across 1101 ZCTA’s were included in the current study. A little over half (54.3%) identified as female and the majority identified as Non-Hispanic White (86.9%), followed by 6.8% Non-Hispanic Black, 4.2% other race, and 2.1% Hispanic. Chronic conditions reported included asthma (13.9%), diabetes (14.9%), heart disease (6.6%), cancer (8.4%), chronic obstructive pulmonary disease (1.92%), and arthritis (1.67%). There was substantial variation in sleep duration and mental health across all ZCTA’s. The prevalence of short sleep (<6 h) was 11.9% and insufficient sleep (<7 h) was 34.7% which is comparable to other state data from BRFSS in previous years (2009) [21]. The mean sleep duration was 6.98 ± 1.50 h. Prevalence of poor mental health (>14 days) was 14.5% with a mean of 4.5 ± 8.7. We present demographic and clinical characteristics for the overall sample in Table 1.

### 3.2. Socioeconomic Deprivation and Sleep Duration

We found a higher prevalence of short sleep in areas with higher social deprivation indices (Figure 1). We examined the unadjusted association between SDI and sleep duration in the first set of linear regression models. The unadjusted association between SDI and sleep duration was statistically significant. Specifically, higher socioeconomic deprivation was associated with shorter sleep duration (β = −0.049, *p* < 0.001). We present the unstandardized coefficient regression coefficient (B), standard error (SE), standardized regression coefficient (β), and coefficient of determination (R^2^) for each model in Table 2.

In the second set of models, we examined socioeconomic deprivation and sleep duration after adjusting for covariates (age, sex at birth, racial identity, and education) using multivariable linear regression. The associations between SDI and sleep duration remained statistically significant after adjusting for covariates (*p* < 0.001) (Table 2).

In the unadjusted logistic regression models, individuals living in the most socially deprived areas had 1.6 times higher odds of short sleep (<6 h) (OR = 1.54, 95% confidence interval [1.34, 1.83]) than those living in less socially deprived areas (Table 3). These associations remained statistically significant after adjusting for covariates (short sleep: aOR = 1.33, 95% confidence interval [1.12, 1.57] (Table 3).

### 3.3. Socioeconomic Deprivation and Mental Health

We found a higher prevalence of poor mental health in areas with higher social deprivation indices. We examined the unadjusted association between SDI mental health in the first set of linear regression models. The unadjusted association between SDI and mental health was statistically significant (*p* < 0.001) (Table 2). Higher socioeconomic deprivation was associated with poorer mental health (β = 0.098, *p* < 0.001).

In the second set of models, we examined socioeconomic deprivation and mental health after adjusting for covariates (age, sex at birth, racial identity, and education) using multivariable linear regression. The associations between SDI and mental health remained statistically significant after adjusting for covariates (*p* < 0.001) (Table 2).

In the unadjusted logistic regression models, individuals living in the most socially deprived areas had 1.5 times higher odds of poor mental health (>14 days) (OR = 1.47, 95% confidence interval [1.26, 1.71] than those living in less socially deprived areas (Table 3). These associations remained statistically significant after adjusting for covariates (poor mental health: aOR = 1.18, 95% confidence interval [1.01, 1.39]).

## 4. Discussion

We investigated the associations among socioeconomic deprivation, sleep duration, and mental health status in a large representative sample of Ohio residents during the first year of COVID-19 pandemic. Individuals from the highest socioeconomically deprived areas had an almost 2-fold increase in odds of short sleep and poor mental health after adjusting for covariates (age, sex at birth, racial identity, education, and body mass index). These findings supplement the growing body of evidence on the longstanding sleep and mental health disparities that disproportionately affect those from socioeconomically disadvantaged areas. Sleep and mental health are dynamically interrelated and are essential to socioemotional development, physical health maintenance, and economic advancement. Thus, our findings highlight additional deleterious effects of structural inequities on individual-level outcomes and the disproportionate impact of COVID-19 on disadvantaged areas.

Close to one-third of the individuals in the study reported insufficient sleep (<7 h) and one-quarter short sleep (<6 h). This finding was consistent with other population-based studies of American adults sleeping less than 7–8 h per night [43,44]. Although sleep time remained unchanged from pre-pandemic, increases in short sleepers have been observed in other studies [45,46]. In an international online survey study (ECLB COVID-19), there was a modest increase in self-reported sleep duration as well as increases in poor sleep quality as measured by the Pittsburgh Sleep Quality Index (4.37 ± 2.71 before the lockdown and 5.32 ± 3.23 after the lockdown) [47]. The public health ramifications of insufficient sleep are under-recognized by society with a conservatively estimated economic burden of USD 107 billion [48]. Higher socioeconomic deprivation was associated with shorter sleep in the present study which is consistent with several studies where individual socioeconomic status was associated with shorter sleep duration or poorer sleep quality [49,50,51,52,53,54,55,56,57]. 

It is essential to recognize the negative ramifications of the COVID-19 pandemic on mental health to mitigate short- and long-term consequences. One in six (14.5%) individuals in the current study experienced poor mental health (>14 days) with a mean of 4.5 ± 8.7 days within a 30-day timeframe. These findings are consistent with other observational studies [33,58,59,60,61,62]. Large disease outbreaks are associated with increased mental health problems due to social disconnection through enforcement of stay-at-home orders, isolation, and unemployment [63]. Additionally, social disconnection either real or perceived is a primary risk factor for a suicidal attempt [64].

We found a higher prevalence of poor mental health in areas with higher social deprivation indices. This result is consistent with the literature where greater neighborhood deprivation was associated with worse mental health [65]. Communities with lower socio-economic status have endured the greatest mental and physical health burden from the pandemic compared to adults with a higher socioeconomic status [1,2,3,4]. These results indicate a critical need to understand the relationship between socioeconomic disadvantage and mental health particularly during a global health pandemic in order to mitigate negative short- and long-term consequences. Poor mental health can have profound ramifications including substance use disorders, drug overdose, and lost productivity [66].

Some important limitations should be considered when interpreting the results of the current study. First, self-reported sleep duration data were used; hence, estimates may be underestimated based on prior research comparing objective sleep duration with self-report [67,68]. People often over-estimate their sleep duration by reporting time in bed versus time asleep [68]. However, in large epidemiological studies, self-reported sleep data have been shown to be reliable [24]. Findings should be corroborated with objective measures of sleep duration to replicate the findings observed in the present study. Second, the use of a cross-sectional design prevents us from investigating changes over time and attributing causality to changes in sleep and mental health trends. Third, respondents may have under- or over-reported mental health symptom frequency leading to increased or decreased estimation of mental health symptom experience in our study. Fourth, our sample was overrepresented in Non-Hispanic White (86.9% vs. 77%), underrepresented in Non-Hispanic Black (6.8% vs. 12.5%), comparable to Asian (0.9% vs. 2.5%), and comparable to Hispanic (2.1% vs. 4.4%) when compared to published 2020 Ohio demographics (census.gov). Finally, there may be societal or cultural variations in how sleep duration and mental health symptoms are reported [69]. Nonetheless, the large sample size with demographic variability contributes to the literature and generalizability of the findings.

The poor sleep and health outcomes observed in Ohio communities with the worst socioeconomic deprivation might be a common phenomenon in the other parts of the country. Timely and geographically targeted public health interventions from local and national government and health care providers are needed to facilitate access to resources and care in historically underserved areas. In addition, public health initiatives aimed at addressing social determinants of health and developing a safety net system are required to mitigate the worsening effects of the pandemic on sleep and mental health.

We encourage replication of our findings in other areas of the United States and in other countries. Sleep disturbance is linked to the occurrence and worsening of mental health problems, which in turn may affect sleep duration and quality. Investigators should explore the bidirectional relationship between sleep and mental health. Additionally, investigators may consider examining populations at a comparable geographic level like ZCTA or to examine larger areas (e.g., state/country) for broader generalizable findings, or smaller clusters (e.g., neighborhoods) to understand conditional processes that can be used for prescriptive multilevel interventions targeting specific domains of socioeconomic deprivation. Our findings underscore the need for geographically targeted public health interventions that support the sleep and mental health of the most socioeconomically disadvantaged. Ultimately, this work can be used to support local, regional, or national policy initiatives to modify societal infrastructures that enable social inequities to perpetuate ongoing healthcare disparities.

## 5. Conclusions

The underlying mechanisms and trends between socioeconomic disadvantage, sleep duration, and mental health and whether these trends vary by sex, gender identity, or racial identity warrant further investigation. Socioeconomic deprivation characteristics may serve as novel targets to improve sleep and mental health in the United States population. In order to reduce these socioeconomic differences, key stakeholders and policymakers should consider the characteristics of the areas in which people live as well as the residents in those areas. The rising trend in short sleep and mental illness warrants further attention. Further, the current study highlights the need to address how catastrophic events disrupt sleep and mental health particularly in socioeconomically disadvantaged areas.

## Figures and Tables

**Figure 1 ijerph-19-14367-f001:**
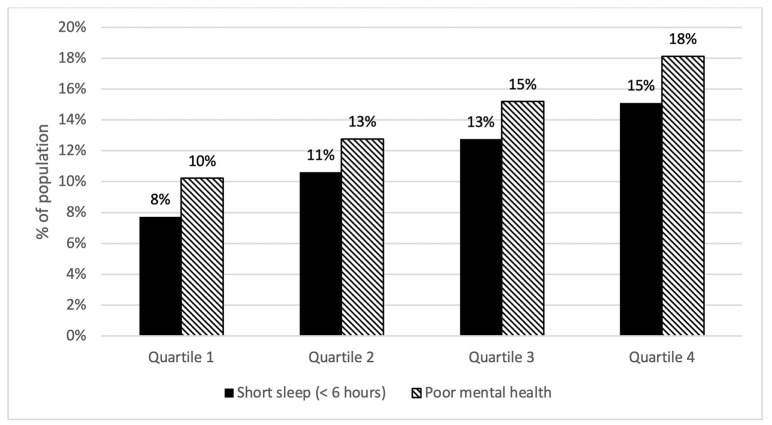
Short sleep and poor mental health prevalence % by social deprivation index quartiles.

**Table 1 ijerph-19-14367-t001:** Demographic Characteristics of the Sample.

Variable	Number (%) or Mean (SD)
Age	54.9 (18.1)
Race/Ethnicity White African American/Black Asian American Indian/Alaskan Native Other Hispanic	12,754 (86.9) 992 (6.8) 139 (0.9) 96 (0.7) 385 (2.6) 310 (2.1)
Sex at Birth Female Male	7970 (54.3) 6706 (45.7)
Years of Education Never Attended or Only Kindergarten Elementary (Grades 1–8) Some Highschool (Grades 9–11) High School Graduate Some College (1–3 years) College Graduate (4 or more years)	6 (0) 165 (1.1) 732 (5) 4932 (33.6) 4086 (27.7) 4723 (32.2)
Annual Household Income Less than USD 10,000 Less than USD 25,000 Less than USD 50,000 Less than USD 75,000 USD 75,000 or more	495 (3.4) 2667 (18.5) 2965 (20.6) 1897 (13.2) 3666 (25.5)
Marital Status Married Divorced/Widowed/Separated/Never Married	7185 (49) 7335 (50)
Body Mass Index	29.1 (6.9)

**Table 2 ijerph-19-14367-t002:** Socioeconomic deprivation and covariates to sleep health and mental health outcomes (Linear Regression Models).

Model	Independent Variable/s	Dependent Variable/s	B	SE	β	*p* Value	R^2^
Model 1	SDI	Sleep Health	−0.003	0.001	−0.049	<0.001	0.002
		Mental Health	0.033	0.003	0.098	<0.001	0.010
Model 2	SDI	Sleep Health	−0.002	0.001	−0.037	<0.001	0.018
		Mental Health	0.022	0.003	0.064	<0.001	0.060

Note. Covariates adjusted for age, sex at birth, racial identity, education, income, and body mass index. SDI = social deprivation index. Model 1 is unadjusted. Model 2 is adjusted for covariates. B is the unstandardized coefficient regression coefficient. SE = standard error. β is the standardized regression coefficient. R^2^ = coefficient of determination shown for each model.

**Table 3 ijerph-19-14367-t003:** Socioeconomic deprivation, sleep health, and mental health (Logistic Regression Models). Residence in 10% most-deprived areas.

Social Deprivation Index						
	OR	95% CI	*p* Value	aOR	95% CI	*p* Value
Short sleep	1.57	[1.34, 1.83]	<0.001	1.33	[1.12, 1.57]	0.001
Poor mental health	1.47	[1.26, 1.71]	<0.001	1.18	[1.01, 1.39]	0.043

Note: CI = confidence interval; OR = odds ratio; aOR = adjusted odds ratio; short sleep < 6 h; poor mental health > 14 poor mental health days. Covariates adjusted for: age, sex at birth, racial identity, education, income, and body mass index.

## Data Availability

Data are available upon request from https://odh.ohio.gov/know-our-programs/behavioral-risk-factor-surveillance-system/data-and-publications (accessed on 1 August 2022) through a data use agreement.

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
