# Peer review of "Socioeconomic Deprivation, Sleep Duration, and Mental Health during the First Year of the COVID-19 Pandemic"

_ijerph, 2022, doi:10.3390/ijerph192114367_

Round 1
Reviewer 1 Report
Very interesting and tremendously relevant topic for potential readers of this Journal.
Very well presented work. Congratulations!
Some comments are made in favor of improving the current version of the manuscript.
.- Abstract. ok.
.- Keywords. Maybe instead of “sleep”, add “sleep duration”. To assess add "socioeconomic deprivation", "health outcomes" and "social deprivation index"
.- Introduction. It turns out long. Consider shortening a paragraph. It is missing in this section if studies have been carried out in this regard (few, many...)
.- Methodology. Doubts arise, when it mentions that the study was approved by the institution of the corresponding author, ¿could the number and type of institutions be briefly explained'; Was the sample size calculated?
.- Results. ok.
.- Discussion. Maybe it's a bit long. It is lacking to contrast more the results with other studies. A reference is made to “other studies” without going into more detail. Last paragraph, avoid repetitions. The terms “researchers” and “clusters” are repeated.
Doubt arises, surprising result on BMI 28.9 (Table 1). Is there any justification for this result? Is there a relationship with the socioeconomic level plus BMI in more economically disadvantaged environments? Or does a higher BMI increase the risk of poorer sleep quality (OSAS or other disorders)?
.- Conclusions. Perhaps the phrase “Our findings add to the existing 351 literature by describing the various ways in which the COVID-19 pandemic has 352 affected sleep and mental health” could be deleted from this section.
.- Tables and Figures. ok.
.- References. Of the 56 limited citations, 24 are recent (equal to or less than 5 years of publication). This represents 43%. To value include some additional recent reference.
Reviewer 2 Report
The paper presents the results of a wide survey investigating the association between socio-economic deprivation, mental health and sleep duration among Ohio residents amidst the Covid-19 pandemic. The study is interesting, well-conceived and relevant, and the sample is large and representative. The background and the rationale for studying the above mentioned relationships are clearly defined and the implication of the observed findings for publich health are thoroughly discussed. Data concerning the association between socio-economic factors and both sleep and mental health are pivotal and the the research design proposed by the authors is definitely appropriate. Overall, the paper is remarkable and can be considered for publication in its present form.
Below, I report a few notes mainly concerning misprints, formatting, and minor issues.
- Reading the text, I assumed that B are unstandardized regression coefficients and Betas are standardized regression coefficients. Although this is a widely known convention, this should be made explicit.
- I assume that p-values in the tables presenting the regression models' results are referrred to the specific coefficients. This could be made clearer in Table 2 by putting the R-squared in the last column and moving the column with the p-values closer to that featuring the regression coefficients.
- Line 218: it seems that a wrong beta coefficient is reported (beta=-.039 instead of .069).
- Line 356: Is "Patents None" intended to be the sixth paragraph heading or was it supposed to appear in the subsequent statements' section?
As a sidenote, I appreciated the authors' acknowledgement of their study's limitations. Although I understand that mantaining the questionnaire compact is a priority in such large survey, I wonder whether they are considering introducing standardized instruments in future studies along with single items (which are indeed well conceived and fundamental). Very short scales are available for both sleep problems and psychological distress, and their usage could further improve the relevance of their results.
